# National trends and resource associated with recurrent penetrating injury

Nam Yong Cho[1], Russyan Mark Mabeza[1], Syed Shahyan Bakhtiyar[1,2], Shannon Richardson[1], Konmal Ali[1], Zachary Tran[1,3], Peyman Benharash[1] *

1 Cardiovascular Outcomes Research Laboratories (CORELAB), David Geffen School of Medicine, University of California, Los Angeles, Los Angeles, CA, United States of America, 2 Department of Surgery, University of Colorado, Aurora, CO, United States of America, 3 Department of Surgery, Loma Linda University Health, Loma Linda, CA, United States of America

* Pbenharash@mednet.ucla.edu

## Abstract

### Background

While recurrent penetrating trauma has been associated with long-term mortality and disability, national data on factors associated with reinjury remain limited. We examined temporal trends, patient characteristics, and resource utilization associated with repeat firearm-related or stab injuries across the US.

### Methods

This was a retrospective study using 2010–2019 Nationwide Readmissions Database (NRD). NRD was queried to identify all hospitalizations for penetrating trauma. Recurrent penetrating injury (RPI) was defined as those returned for a subsequent penetrating injury within 60 days. We quantified injury severity using the International Classification of Diseases Trauma Mortality Prediction model. Trends in RPI, length of stay (LOS), hospitalization costs, and rate of non-home discharge were then analyzed. Multivariable regression models were developed to assess the association of RPI with outcomes of interest.

### Results

Of an estimated 968,717 patients (28.4% Gunshot, 71.6% Stab), 2.1% experienced RPI within 60 days of the initial injury. From 2010 to 2019, recurrent gunshot wounds increased in annual incidence while that of stab cohort remained stable. Patients experiencing recurrent gunshot wounds were more often male (88.9 vs 87.0%, P<0.001), younger (30 [23–40] vs 32 [24–44] years, P<0.001), and less commonly insured by Medicare (6.5 vs 11.2%, P<0.001) compared to others. Those with recurrent stab wounds were younger (36 [27–49] vs 44 [30–57] years, P<0.001), less commonly insured by Medicare (21.3 vs 29.3%, P<0.001), and had lower Elixhauser Index Comorbidities score (2 [1–3] vs 3 [1–4], P<0.001) compared to others. After risk adjustment, RPI of both gunshot and stab was associated with significantly higher hospitalization costs, a shorter time before readmission, and increased odds of non-home discharge.

**Data Availability Statement:** All relevant data are within the manuscript and its Supporting Information files.

**Funding:** The authors received no specific funding for this work.

**Competing interests:** The authors have declared that no competing interests exist.

## Conclusion

The trend in RPI has been on the rise for the past decade. National efforts to improve post-discharge prevention and social support services for patients with penetrating trauma are warranted and may reduce the burden of RPI.

## 1. Introduction

Traumatic injuries account for an estimated 1.4 million annual emergency department visits and nearly $8.5 billion in healthcare expenditures [1]. Approximately 44% of these admissions are for recurrent traumatic injuries in patients with prior injury-related hospitalization. Primarily consisting of gunshot and stab injuries, recurrent penetrating injury (RPI) has been linked to increased long-term mortality [2], with 20% of patients facing death within five years of a repeat injury. Furthermore, individuals with RPI are likely to present with a second unrelated trauma or death. Such patterns present a critical public health challenge, indicating a need for a thorough analysis of factors associated with RPI.

Previous institutional studies have noted younger age, Black race, and low-income status to be risk factors for RPI [3]. Furthermore, individuals suffering from recurrent penetrating trauma frequently suffer from high morbidity and costs of care due to often required operative interventions and extended inpatient stays. Despite being a significant issue, examination of RPI remains limited to single-center series. Thus, the present study examined national trends as well as factors associated with RPI. The aim of this study was to elucidate clinical and financial outcomes following RPI using a nationally representative cohort.

## 2. Material and methods

### 2.1 Data source and study cohort

This was a retrospective study using 2010–2019 Nationwide Readmissions Database (NRD). Maintained by the Health Care Costs and Utilization Project (HCUP), the NRD is the largest national readmission database in the US and provides accurate national estimates for nearly 60% of all annual hospitalizations. The NRD contains unique identifiers for the patient and hospital variables, which allows readmissions to be tracked within each calendar year. Due to the de-identified nature of data, this study was deemed exempt from the Institutional Review Board at the University of California, Los Angeles.

All adult ($\geq$18 years) hospitalizations for gunshot- and stab-related trauma were identified using the *External Cause of Injury* (ECodes) and *International Classification of Diseases*, *Tenth Revision* (ICD-10) codes. Entries missing key information, such as sex, age or mortality, were excluded from the analysis. We quantified injury severity using the *International Classification of Diseases* Trauma Mortality Prediction Model [4]. The cohort was stratified by gunshot and stab injuries. In each group, patients were further classified as gunshot (*GSW-R*) and stab (*Stab-R*) recurrent injuries. RPI was defined as a subsequent penetrating injury requiring readmission within 60 days of discharge from the index hospitalization. The timing of 60 days was used to capture at least the 75th percentile of patients readmitted with RPI while minimizing the skewed distribution.

## 2.2 Study variables and outcomes

Patient and hospital characteristics such as age, insurance status, the intent of the injury, hospital region, and income quartiles were defined according to the NRD data dictionary [5]. The Elixhauser Comorbidity Index, a validated composite score of 30 chronic comorbidities [6], was used to estimate the burden of comorbidities in the study population. In addition, specific patient comorbidities were further ascertained using ICD-9/10 codes, including congestive heart failure, diabetes, and hypertension, among others (S1 Table). Mortality analysis was limited to death during hospitalization, whereas non-home discharge was defined as transfer to a post-acute care facility. Hospitalization costs were calculated by applying center-specific cost-to-charge ratios to total hospitalization charges and adjusted for inflation using the 2019 Bureau of Labor Statistics Personal Health Care Price Index. The primary outcome of interest was the trends in RPI, while secondary outcomes included index hospitalization length of stay (LOS), cumulative hospitalization costs, and non-home discharge.

## 2.3 Statistical methods

Categorical variables are reported as percentages, and continuous variables as medians with interquartile range (IQR). The Chi-square and Kruskal-Wallis tests were used to compare patient characteristics in both study cohorts. Temporal trends were assessed using a nonparametric test (nptrend). Multivariable regressions were used to identify patient and hospital factors associated with RPI as well as secondary outcomes. We used Elastic Net regularization to guide variable selection to improve out-of-sample generalizability [7]. Regression outputs are reported as adjusted odds ratios (AOR) or beta coefficients (β) with 95% confidence intervals (95% CI). A P-value <0.05 was considered statistically significant. All statistical analyses were performed using Stata 16.1 (StataCorp, College Station, TX).

## 3. Results

### 3.1 Demographics of patients and trends of GSW-R

Of an estimated 280,854 patients, 5,588 (2.0%) experienced RPI within 60 days of the initial injury (Fig 1). From 2010 to 2019, hospitalizations due to gunshot wound injury increased by 138% and recurrent GSW by 196% in annual incidence (nptrend<0.001, Fig 2). Additionally, non-home discharge among *GSW-R* increased from 16.8% to 20.2% over the study period (nptrend<0.001, Fig 3). Compared to those without recurrent injury, patients in the *GSW-R* cohort were more commonly male (88.9 vs 87.0%, P<0.001), younger (30 [23–40] vs 32 [24–44] years, P<0.001), and insured by Medicaid (36.3 vs 32.1%, P<0.001). In addition, patients with RPI more commonly experienced gunshot injury due to accidents (37.2 vs 32.7%, P<0.001) compared to others. Elixhauser comorbidity Index score and prevalence of psychiatric disorders were similar between those with RPI and others (Table 1).

### 3.2 Outcomes associated with GSW-R

Patients with recurrent GSW had shorter unadjusted index LOS (5 [2–10] vs 8 [4–18] days, P<0.001) and lower median index hospitalization cost ($18,000 [8,800–37,600] vs $28,800 [14,500–61,100], P<0.001), relative to others. Compared to those without RPI, *GSW-R* had higher rates of non-home discharge (30.6 vs 25.9%, P<0.001). After risk adjustment, RPI was associated with significantly higher cumulative hospitalization costs (β $19,200, 95% Confidence Interval [CI] $15,800-$22,300), a shorter estimated LOS by 0.7 days (95%CI 0.1–1.3 days), and increased odds of discharge to a short-term care facility (Adjusted Odds Ratio [AOR] 5.4, 95%CI 4.3–6.8, Table 3). As shown in Fig 4, comorbidities including

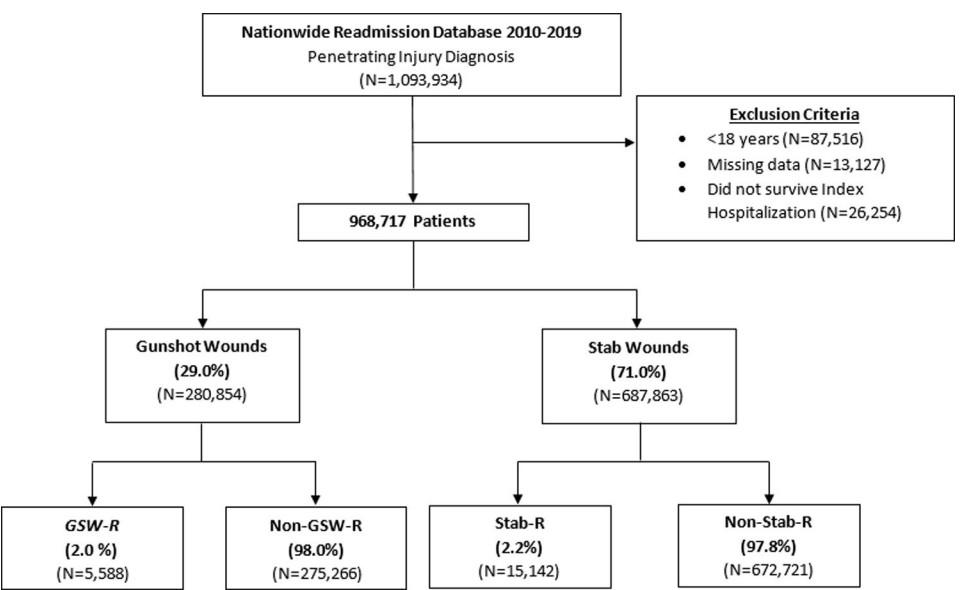

**Fig 1. CONSORT (Consolidated Standards of Reporting Trials) diagram of study cohort and survey-weighted sample size.** GSW-R, Gunshot recurrent injury. Stab-R, Stab recurrent injury.

hypothyroidism, psychoses, and opioid use disorder were not associated with increased odds of recurrent gunshot injury.

### 3.3 Demographics of patients and trends of stab injury-R

Of an estimated 687,863 patients, 15,142 (2.2%) were identified as *Stab-R* cohort. Over the study period, there was a stable trend in hospitalization due to recurrent stab injury (nptrend = 0.60, Fig 2). Similarly, non-home disposition among recurrent stab wound patients increased over the study period from 14.4% to 15.4% (nptrend <0.001, Fig 3). Compared to

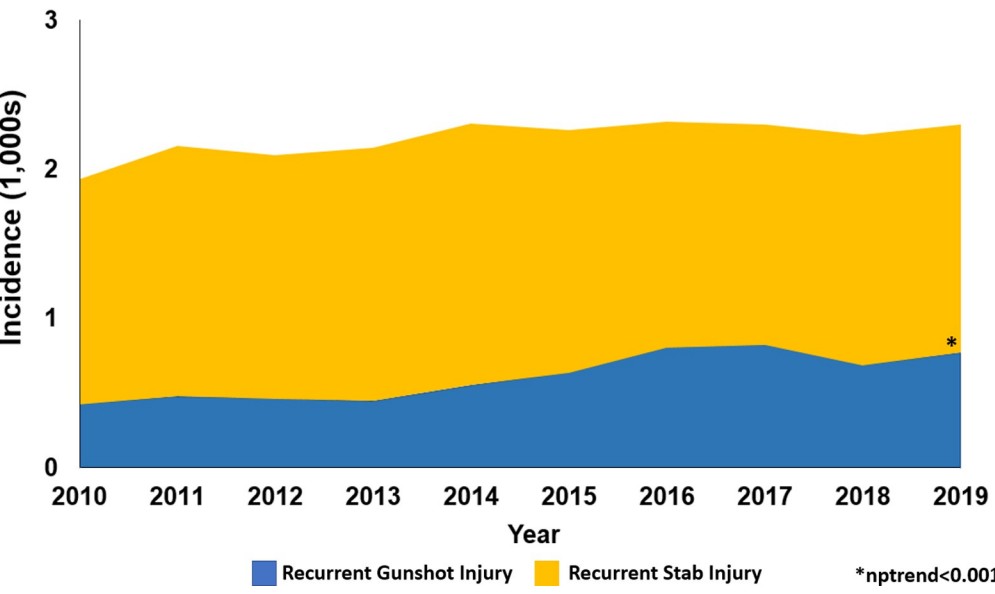

**Fig 2. National trends in recurrent penetrating injuries from 2010 to 2019.**

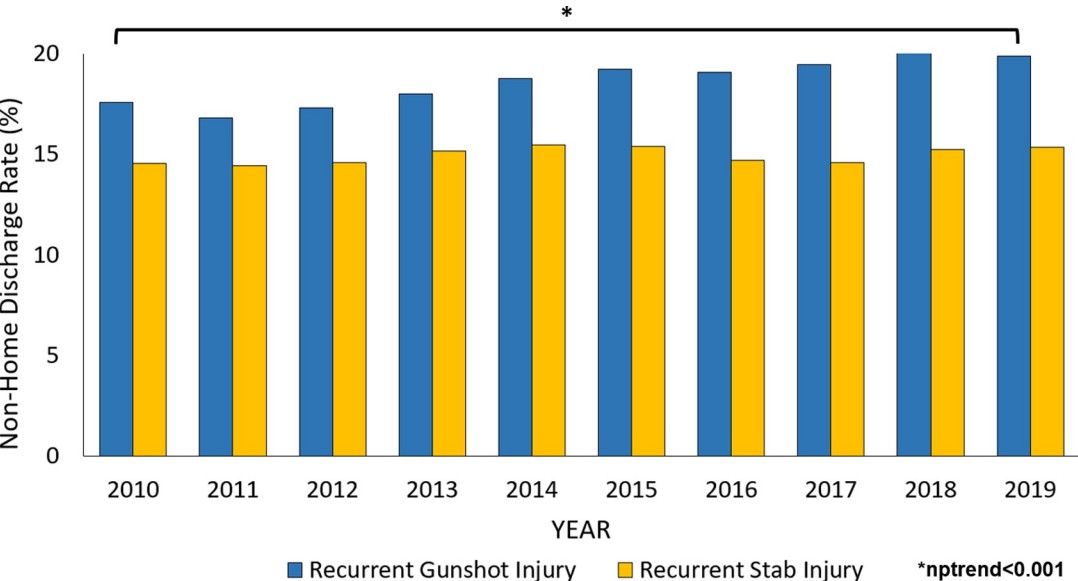

**Fig 3. National trends in non-home discharge for patients experiencing recurrent penetrating trauma from 2010 to 2019.**

those without repeated injury, patients from *Stab-R* were younger (36 [27–49] vs 44 [30–57] years, P<0.001), more often had an injury due to self-harm (65.8 vs 50.7%, P<0.001) compared to others. Furthermore, patients with recurrent stab injury had higher rates of depression (47.1 vs 30.1%, P<0.001), alcohol use disorder (24.7 vs 19.6%, P<0.001), and psychoses (18.4 vs 7.5%, P<0.001) relative to those without recurrent injuries (Table 1).

### 3.4 Outcomes associated with *Stab-R*

As shown in Table 2, underinsured and low-income status were associated with *Stab-R*. Those from *Stab-R* had shorter unadjusted index LOS (4 [2–7] vs 5 [3–8] days, P<0.001) and lower median index hospitalization cost ($6,600 [3,600–12,800] vs $7,900 [4,200–15,900], P<0.001) compared to others. Additionally, those with recurrent stab injuries had higher rates of non-home discharge at index hospitalization (21.8 vs 17.3%, P<0.001) compared to others. After risk adjustment, *Stab-R* was associated with significantly higher cumulative hospitalization costs (β $13,300, 95% Confidence Interval [CI] $11,600-$15,000), shorter estimated LOS by 0.4 days (95%CI 0.4–0.6 days), and increased odds of transfer to a short-term care facility (AOR 2.9, 95%CI 2.4–3.6) compared to home disposition (Table 3). Comorbid conditions of hypothyroidism (AOR 1.36, 95%CI 1.18–1.58) and psychoses (AOR 1.41, 95%CI 1.28–1.56) were associated with a greater likelihood of *Stab-R* (Fig 4).

## 4. Discussion

Penetrating trauma has been associated with recurrence and is responsible for >$25 billion in annual costs based on medical expenses and lost productivity in the US [8]. Furthermore, the recurrence of gunshot and stab injuries underscores the prevalence of high-risk factors in a subset of the population at the highest risk for traumatic injuries in the United States. Previous studies have shown that RPI is likely not a random event but instead an indicator of elevated risk for excessive violence-related behaviors and exposures in the population [1, 3]. Importantly, patients presenting with recurrent penetrating trauma have significantly prolonged complications resulting from the injury and have a 77% higher likelihood of long-term

**Table 1. Demographic and clinical characteristics of patients with recurrent gunshot and stab injuries.**

| | Recurrent Gunshot Injury | | | Recurrent Stab Injury | | |
|---|---|---|---|---|---|---|
| | GSW-R (N = 5,588) | No GSW-R (N = 275,266) | P-Value | Stab-R (N = 15,142) | No Stab-R (N = 672,721) | P-Value |
| Female (%) | 609 (10.9) | 33,032 (12.0) | 0.17 | 5,860 (38.7) | 197,107 (29.3) | <0.001 |
| Age, years | 30 [23–40] | 32 [24–44] | <0.001 | 36 [26–47] | 39 [27–53] | <0.001 |
| Elixhauser Index | 1 [0–2] | 1 [0–2] | 0.98 | 2 [1–3] | 1 [0–3] | <0.001 |
| TMPM Score | 0.02 [0.01–0.21] | 0.02 [0.01–0.20] | 0.58 | 0.01 [0.01–0.02] | 0.02 [0.01–0.03] | <0.001 |
| Hospital Bed Size, (%) | | | 0.99 | | | 0.86 |
| Small | 335 (6.0) | 15,690 (5.7) | | 1,484 (9.8) | 67,272 (10.0) | |
| Medium | 1,313 (23.5) | 65,789 (23.9) | | 3,574 (23.6) | 158,089 (23.5) | |
| Large | 3,940 (70.5) | 193,787 (70.4) | | 10,085 (66.6) | 447,359 (66.5) | |
| Hospital Teach Status, (%) | | | 0.81 | | | 0.03 |
| Rural | 626 (11.2) | 31,380 (11.4) | | 4,073 (26.9) | 168,853 (25.1) | |
| Metropolitan Nonteaching | 4,805 (86.0) | 235,353 (85.5) | | 9,782 (64.6) | 446,014 (66.3) | |
| Metropolitan Teaching | 156 (2.8) | 8,533 (3.1) | | 1,287 (8.5) | 57,181 (8.5) | |
| Insurance Status, (%) | | | <0.001 | | | <0.001 |
| Medicare | 363 (6.5) | 20,645 (7.5) | | 3,150 (20.8) | 116,381 (17.3) | |
| Medicaid | 2,028 (36.3) | 88,360 (32.1) | | 4,845 (32.0) | 166,836 (24.8) | |
| Private | 1,123 (20.1) | 56,430 (20.5) | | 2,983 (19.7) | 161,453 (24.0) | |
| Self-pay | 1,347 (24.1) | 71,569 (26.0) | | 1,741 (11.5) | 116,381 (17.3) | |
| Income quartile, (%) | | | 0.75 | | | 0.02 |
| 0 - 25th | 2,945 (52.7) | 146,442 (53.2) | | 5,542 (36.6) | 244,198 (36.3) | |
| 25th - 50th | 1,291 (23.1) | 64,412 (23.4) | | 4,225 (27.9) | 180,289 (26.8) | |
| 50th - 75th | 939 (16.8) | 43,767 (15.9) | | 3,346 (22.1) | 147,326 (21.9) | |
| 75th - 100th | 413 (7.4) | 20,645 (7.5) | | 2,029 (13.4) | 101,581 (15.1) | |
| Injury Intent | | | <0.001 | | | <0.001 |
| Assault | 3,062 (54.8) | 150,846 (54.8) | | 1,287 (8.5) | 123,781 (18.4) | |
| Accident | 2,073 (37.1) | 92,214 (33.5) | | 3,437 (22.7) | 306,088 (45.5) | |
| Legal | 101 (1.8) | 4,955 (1.8) | | 3 (0) | 306 (0) | |
| Self-Harm | 352 (6.3) | 27,251 (9.9) | | 10,415 (68.8) | 242,546 (36.1) | |
| Comorbidities | | | | | | |
| Hypothyroidism | 56 (1.0) | 2,477 (0.9) | 0.91 | 1,181 (7.8) | 30,272 (4.5) | <0.001 |
| OUD | 140 (2.5) | 6,882 (2.5) | 0.99 | 1,196 (7.9) | 34,981 (5.2) | <0.001 |
| Psychoses | 1,177 (2.1) | 5,781 (2.1) | 0.92 | 2,786 (18.4) | 50,454 (7.5) | <0.001 |

GSW-R, Gunshot recurrent injury. Stab-R, Stab Recurrent injury. OUD, opioid use disorder. TMPM, Trauma Mortality Prediction Model. TMPM score indicates predicted probability of death based on patient's five most severe injuries.

mortality following RPI [2]. Nonetheless, data regarding trends, LOS, hospitalization expenditure, non-home discharge rate, and factors associated with RPI is lacking in the literature. In the present work, we noted an increasing national trend in recurrent gunshot injury, with nearly 2.1% experiencing this event within 60 days of initial injury. Over the study period, the number of hospitalization due to recurrent stab injuries remained similar. We also found patients with a non-home disposition to face increased odds of re-injury. Several of these findings warrant further discussion.

Consistent with a rise in firearm-related morbidity and mortality in the US, we noted a significant escalation of recurrent gunshot injury over the study period. Non-fatal GSWs are two-fold more common than fatal injuries [9], and surviving victims are predisposed to a cycle of violence. Among patients experiencing recurrent gunshot injury, the present study showed

**Table 2. Risk-adjusted outcomes in patients with recurrent penetrating trauma.**

| | Recurrent Gunshot Injury | | Recurrent Stab Injury | |
|---|---|---|---|---|
| | AOR | 95% CI | AOR | 95% CI |
| Female Sex | 0.96 | 0.81–1.13 | 0.93 | 0.86–1.01 |
| Age, (per years) | 0.99 | 0.99–1.00 | 0.99 | 0.99–0.99 |
| Elixhauser Index Score | 1.07* | 1.01–1.13 | 1.03 | 0.99–1.07 |
| Hospital Bed Size, (%) | | | | |
| Small | REF | | REF | |
| Medium | 0.94 | 0.83–1.07 | 0.98 | 0.96–1.14 |
| Large | 0.84* | 0.66–1.06 | 0.87* | 0.76–0.98 |
| Hospital Teach Status, (%) | | | | |
| Rural | REF | | REF | |
| Metropolitan Nonteaching | 1.03 | 0.86–1.23 | 1.04 | 0.96–1.14 |
| Metropolitan Teaching | 0.79 | 0.75–1.20 | 0.82* | 0.69–0.97 |
| Insurance Status, (%) | | | | |
| Medicare | REF | | REF | |
| Medicaid | 1.30 | 0.99–1.69 | 0.85* | 0.76–0.95 |
| Private | 1.21 | 0.92–1.58 | 0.59* | 0.53–0.68 |
| Self-pay | 1.30 | 0.99–1.71 | 0.56* | 0.48–0.65 |
| Income quartile, (%) | | | | |
| 0 - 25th | REF | | REF | |
| 25th - 50th | 1.00 | 0.88–1.14 | 1.00 | 0.92–1.10 |
| 50th - 75th | 1.09 | 0.94–1.26 | 0.95 | 0.86–1.04 |
| 75th - 100th | 1.10 | 0.90–1.33 | 0.83* | 0.74–0.93 |
| Comorbidities, (%) | | | | |
| Hypothyroidism | 0.76 | 0.42–1.37 | 1.36* | 1.18–1.58 |
| OUD | 1.04 | 0.69–1.28 | 1.01 | 0.88–1.15 |
| Psychoses | 1.07 | 0.77–1.49 | 1.41* | 1.23–1.52 |
| Injury Intent, (%) | | | | |
| Assault | REF | | REF | |
| Accident | 1.06 | 0.95–1.19 | 1.05 | 0.91–1.21 |
| Legal | 1.11 | 0.75–1.64 | 2.50 | 0.61–10.2 |
| Self-Harm | 0.96 | 0.75–1.24 | 4.1* | 3.58–4.72 |

GSW-R, recurrent gunshot injury. Stab-R, recurrent stab injury. AOR, Adjusted Odds Ratio. CI, Confidence Interval. REF, Reference. OUD, Opioid Use Disorder.

that 91.8% of initial gunshot wounds occur due to assault or accident. These findings highlight the prevalence of urban violence resulting from inadequate firearm regulation. Policies to promote gun safety and laws that mitigate gun-related danger in communities are necessitated to prevent recurrent gun injury in high-risk patients. Moreover, the lack of gun-related injury prevention efforts among perioperative physicians has also been proposed as an area of improvement [10, 11]. Healthcare professionals are frequently the first or only caregivers to come in contact with victims of firearm-related injury. As such, it is essential for physicians to identify patients at risk for RPI. Holistic interventions involving physicians controlling substance use, providing psychiatric care for mood disorders, and facilitating prevention measures may help reduce recurrent gunshot injury.

In contrast to recurrent gunshot injury, the respective rates for stab injury have remained steady over the past decade. Interestingly, we found a significant association between *STAB_R*

**Table 3. Risk-adjusted outcomes in patients with recurrent gunshot and stab injuries.**

| | Recurrent Gunshot Injury | | Recurrent Stab Injury | |
|---|---|---|---|---|
| | Estimates | 95% CI | Estimates | 95% CI |
| LOS, (d) | -0.7 | -1.3–0.1 | -0.4 | -0.6–0.4 |
| Cumulative Cost, ($1,000) | 19.1 | 15.8–22.3 | 13.3 | 11.6–15.0 |
| Discharge Location | | | | |
| Home | Reference | | Reference | |
| Short-Term Care Facility | 5.4 | 4.3–6.8 | 2.9 | 2.4–3.6 |
| Skilled Nursing Facility | 1.3 | 1.0–1.6 | 1.0 | 0.9–1.1 |
| Home Health | 1.5 | 1.3–1.7 | 1.2 | 0.9–1.4 |
| Against Medical Advise | 3.0 | 2.4–3.8 | 1.9 | 1.6–2.3 |

Estimates are reported as AORs or β-coefficients for binary and continuous variables, respectively. CI; Confidence Interval; d; days.

and comorbid conditions such as psychoses and hypothyroidism. We further noted that 68.8% of patients experiencing recurrent stab injuries were injured due to self-harm. Self-injurious behaviors are strongly associated with mental disorders, including psychosis and depression [12, 13]. Borde and colleagues have shown increased odds of clinical depression with any severity of hypothyroidism [14]. In addition to clinical depression, dysregulation of thyroid function can manifest with a wide range of psychiatric symptoms [15], including altered personality with psychotic symptoms, both of which can ultimately lead to self-harming behaviors. These findings highlight potentially modifiable factors that increase the risk for recurrent stab injuries. It is also important to highlight low socioeconomic status and a prolonged period of untreated psychiatric illness as contributing to the risk of self-harm [16]. In the present study, we noted Medicare or Medicaid coverage and low-income status to be risk factors for recurrent stab injury. Although the exact nature of the association remains unclear, it is not unreasonable to assume that underserved patients lack longitudinal care that may mitigate high-risk behaviors or urban trauma exposure. Easier provider access to screening tools or

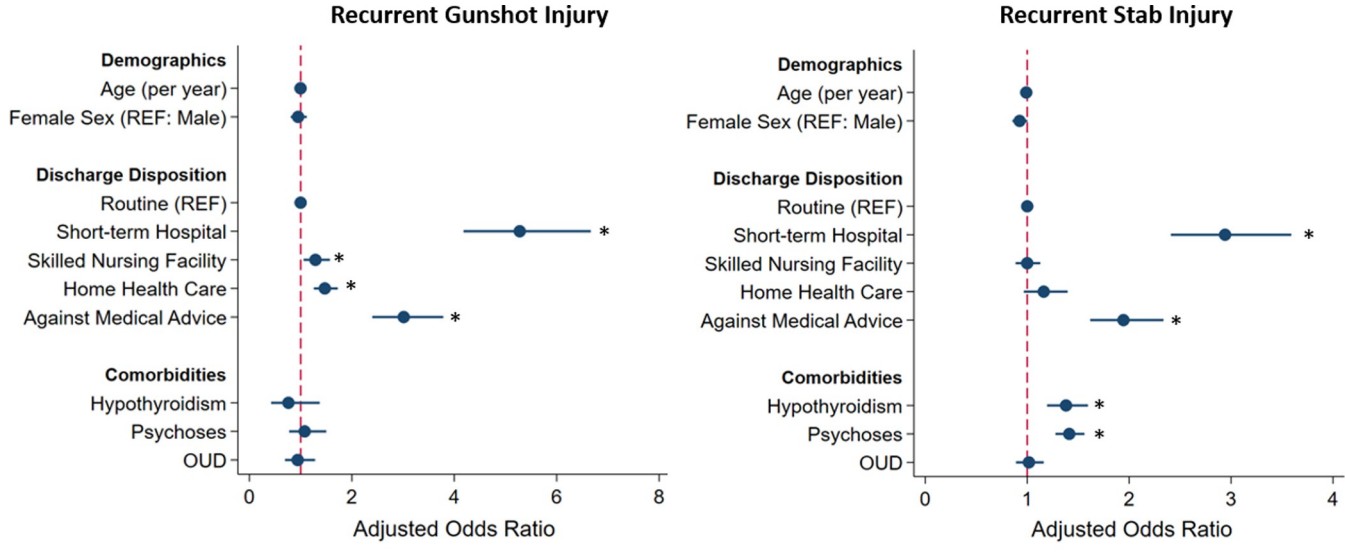

**Fig 4. Risk-adjusted predictors of recurrent gunshot and stab injuries. REF; reference. OUD, Opioid Use Disorder.**

adoption of a trauma-informed approach may reduce RPI in patients with concomitant psychiatric illnesses or inadequate access to resources.

Regardless of the primary penetrating trauma, we found non-home discharge to be associated with higher rates of RPI. Leaving against medical advice can limit appropriate rehabilitation, psychiatric evaluation, and social services in patients with a traumatic injury. Ibrahim et al. showed younger age, Black race, low household income, and lack of medical insurance as predictors of being discharged against medical advice [17]. These factors align with risk factors for trauma and a lower likelihood of receiving follow-up care and rehabilitation [18]. Prior direct or indirect violence exposure is highly prevalent among patients experiencing RPI [19], often intensifying the post-traumatic stress symptoms of sleep disturbance and depression. Being transferred to a short-term medical facility restrains a proper psychiatric evaluation and biopsychosocial treatment in patients, which may be necessary to prevent self-harming behaviors. Treatment for penetrating injuries at trauma centers having expertise in the management of such patients and scheduling adequate follow-up care may reduce this RPI [20]. Nevertheless, substantial geographical locations and variations in emergency medical service availability often hinder access to trauma center services [21], leaving patients with limited options for discharge. Taken into the context of risk for RPI, institutional programs to provide longitudinal post-trauma monitoring and treatment may benefit patients following penetrating trauma. In addition, intervention and prevention programs targeting gun violence in urban and rural areas are warranted to reduce recurrent injuries among trauma patients.

This study had several important limitations due to its retrospective design and the nature of the NRD. Although the AHRQ has quality control measures to ensure best practices for coding, the administrative nature of the NRD allows the potential for miscoded events and missing observations within the database. Additionally, the NRD lacks patient-identifying variables, such as race, location, and hospital data, as well as surgeon-specific variables. The NRD also lacks granular details of readmission between states or calendar years, medication use, and the presence of a pain team consult. Due to our reliance on ICD-10 diagnosis and coding practices at various institutions to define penetrating injuries, patients readmitted due to squalene or complications of their index admissions could not be delineated. Despite the strength of NRD to track repeat hospitalizations at participating hospitals, information regarding the outpatient treatment of RPI as well as out-of-hospital mortality are unable to be determined. We were unable to ascertain affiliations between care facilities and reasons for non-home discharge dispositions. Lastly, hospital-level analysis was limited as we were unable to evaluate trauma center designation. Despite these limitations, we utilized statistically robust methodologies to reduce bias and evaluated RPI using a nationally representative cohort. Our findings highlight multiple factors that may supplement the prevention interventions in high-risk patients of RPI.

The rate of RPI among patients with penetrating trauma has increased over the past decade and continues to pose a clinical and financial burden in the US. Our study underscores the contributing factors underlying repetitive incidences of gunshot and stab injuries. In light of these findings, national efforts to improve post-discharge preventive measures and bolster social support services for patients with penetrating trauma are warranted. To effectively address the mounting burden of RPI, multidisciplinary approaches involving physicians, public health leaders, and legislators should be adopted to mitigate the incidence of penetrating trauma within our communities.

## Supporting information

**S1 Table. International Classification of Disease, Ninth and Tenth revision, codes (ICD-9/10) for penetrating traumas.**
(DOCX)

## Acknowledgments

**Author disclosure statement**

Dr. Peyman Benharash received proctor fees from Atricure as a surgical proctor. This manuscript does not discuss any Atricure products or services. Other authors report no conflicts.

**Meeting presentation**

Accepted for *Oral Presentation* at the 18th Annual Academic Surgical Congress.

Abstract #: ASC20230411

Session: 37 –Clinical/Outcomes: Trauma/Critical Care Oral Session I

Date/Location: Wednesday, February 8, 2023 / Houston, Texas

## Author Contributions

**Conceptualization:** Nam Yong Cho, Syed Shahyan Bakhtiyar, Zachary Tran, Peyman Benharash.

**Data curation:** Nam Yong Cho, Konmal Ali, Peyman Benharash.

**Formal analysis:** Nam Yong Cho, Peyman Benharash.

**Methodology:** Nam Yong Cho.

**Supervision:** Peyman Benharash.

**Validation:** Russyan Mark Mabeza, Syed Shahyan Bakhtiyar, Zachary Tran.

**Writing – original draft:** Nam Yong Cho.

**Writing – review & editing:** Russyan Mark Mabeza, Syed Shahyan Bakhtiyar, Shannon Richardson, Konmal Ali, Zachary Tran, Peyman Benharash.

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
