## [Decision Letter · Decision Letter 0]

10 May 2023

PONE-D-23-00328National Trends and Resource Utilization Associated with Penetrating Trauma RecidivismPLOS ONE

Dear Dr. Benharash,

Thank you for submitting your manuscript to PLOS ONE. After careful consideration, we feel that it has merit but does not fully meet PLOS ONE’s publication criteria as it currently stands. Therefore, we invite you to submit a revised version of the manuscript that addresses the points raised during the review process.

We look forward to receiving your revised manuscript.

Kind regards,

Axel Benhamed, M.D, MSc

Academic Editor

PLOS ONE

Journal Requirements:

Additional Editor Comments: 

Dear Dr Benharash,

Thank you for submitting your manuscript to PLOS ONE. After careful consideration, we feel that it has merit but does not fully meet PLOS ONE’s publication criteria as it currently stands. Therefore, we invite you to submit a revised version of the manuscript that addresses the points raised during the review process.

GENERAL COMMENTS

- First, although the term “trauma recidivism” is common, I would caution the authors against its use. The term “recidivism” is strongly associated with criminality and implies fault on the part of the patient. Patients injured due to violence frequently come from marginalized groups, and this language risks reinforcing racist assumptions about this patient population. Recurrent violent injury is a clear phrase that can be substituted. I recommend authors to use “recurrent traumatic injury” instead.

- Please follow the STROBE guidelines style, especially for the methods section

-

ABTRASCT:

- Results: please provide numerical results

INTRODUCTION:

- Introduction, line 1: What does “urban trauma” mean? Is this study limited to urban areas?

- I recommend that the aim of the study be explicitly stated at the end of the introduction.

METHODS

- How were new injuries distinguished from patients readmitted due to sequelae or complications of their index admission?

- How was the index admission identified? Could patients have been injured prior to the first visit that is denoted here?

- The study uses ICD-9/10 codes to identify firearm- and stab-related trauma, which may not capture all cases or accurately differentiate between intentional and unintentional injuries. How did authors handle this?

- Proportion of missing data, figure 1, supplemental Table 1 are results and should not be mentioned in the methods section but in the results section instead.

RESULTS

- Figure 1: please indicate how many patients were excluded for each exclusion criteria.

- Please provide n and % in tables not only %

- Please indicate the evolution of % GSW between 2010 and 2019

- Paragraph 3.3: Of an estimated 687,863 patients, 15,142 (2.2%) were (and not was) identified.

DISCUSSION

- Discussion, line 1: “Penetrating trauma is the strongest predictor of future traumatic injury.” This is an often-repeated saying, but to my knowledge it has not been demonstrated. The cited reference here simply refers to Sims’ 1989 paper, which does not establish this relationship.

- By using the NRD, this analysis is limited to injuries that result in hospitalization. This is less than half of firearm injuries, with the majority being treated and released from the ED. If this data source is to be used, all of the results and conclusions must be interpreted in this context.

- I believe the NRD does not include race. Given the role of structural racism in generating risk for violent injury, this is a limitation that should be mentioned.

- - 30% of readmissions were very soon after discharge. Again, this makes me suspect that these are either transfers or complications, rather than real new injuries. Please clarify this point

- - Patients discharged to rehab, etc, had higher risk of readmission. Is it really plausible that these were new injuries? Patients with injuries that are severe enough to require rehab are less likely to be out and about. And indeed, since much of the 60 day follow up period would be spent in rehab, I suspect this may be misattribution.

- The conclusion of the study should be appropriately restrained and aligned with its objective

Reviewers' comments:

Reviewer's Responses to Questions

**Comments to the Author**

1. Is the manuscript technically sound, and do the data support the conclusions?

Reviewer #1: Yes

Reviewer #2: Yes

Reviewer #3: Yes

2. Has the statistical analysis been performed appropriately and rigorously? 

Reviewer #1: Yes

Reviewer #2: Yes

Reviewer #3: Yes

3. Have the authors made all data underlying the findings in their manuscript fully available?

Reviewer #1: No

Reviewer #2: Yes

Reviewer #3: No

4. Is the manuscript presented in an intelligible fashion and written in standard English?

Reviewer #1: Yes

Reviewer #2: Yes

Reviewer #3: Yes

5. Review Comments to the Author

Reviewer #1: This paper uses the nationwide readmissions database to seek out patients with an index admission for gunshot wound or stabbing and to track for repeat admissions for new injuries. While I agree that recurrent injury is an important topic that is not fully understood, I have some concerns about the current analysis.

- First, although the term “trauma recidivism” is common, I would caution the authors against its use. The term “recidivism” is strongly associated with criminality and implies fault on the part of the patient. Patients injured due to violence frequently come from marginalized groups, and this language risks reinforcing racist assumptions about this patient population. Recurrent violent injury is a clear phrase that can be substituted.

- By using the NRD, this analysis is limited to injuries that result in hospitalization. This is less than half of firearm injuries, with the majority being treated and released from the ED. If this data source is to be used, all of the results and conclusions must be interpreted in this context.

- How were new injuries distinguished from patients readmitted due to sequelae or complications of their index admission?

- 30% of readmissions were very soon after discharge. Again, this makes me suspect that these are either transfers or complications, rather than real new injuries.

- Patients discharged to rehab, etc, had higher risk of readmission. Is it really plausible that these were new injuries? Patients with injuries that are severe enough to require rehab are less likely to be out and about. And indeed, since much of the 60 day follow up period would be spent in rehab, I suspect this may be misattribution.

- How was the index admission identified? Could patients have been injured prior to the first visit that is denoted here?

- I believe the NRD does not include race. Given the role of structural racism in generating risk for violent injury, this is a limitation that should be mentioned.

- Introduction, line 1: What does “urban trauma” mean? Is this study limited to urban areas?

- Discussion, line 1: “Penetrating trauma is the strongest predictor of future traumatic injury.” This is an often-repeated saying, but to my knowledge it has not been demonstrated. The cited reference here simply refers to Sims’ 1989 paper, which does not establish this relationship

Reviewer #2: Thank you for the opportunity to review the article by Nam Yong Cho ‘National Trends and Resource Utilization Associated with Penetrating Trauma Recidivism’. The article is investigating an important health care issue. Penetrating Trauma Recidivism is a surrogate or ‘the tip of the iceberg’ of a major health care issue in the US: Trauma due to GSW and SW. Therefore, the article is of great importance. The methods are sound and the results are clearly stated. Moreover, an adequate limitations section is included and the conclusions were made cautiously.

Reviewer #3: Thank you for the opportunity to review the manuscript. The aim of this study was to examine temporal trends, patient characteristics, and resource utilization associated with repeat firearm-related or stab injury across the US.

Introduction:

The introduction provides a brief overview of the importance of firearm-related and stab injury across the US.

Material & Methods:

The approach is well described. Also, the statistical analysis is defined.

Results:

The results are presented clearly and briefly. Both the tables and figures are well presented.

Discussion:

The discussion is well written and addresses all relevant results. The limitations in particular are adequately addressed. The conclusion based on the presented results is understandable and appropriate.

6. PLOS authors have the option to publish the peer review history of their article (what does this mean?). If published, this will include your full peer review and any attached files.

Reviewer #1: No

Reviewer #2: **Yes: **Prof. Dr. med. Beat Schnüriger

Reviewer #3: **Yes: **Ass. Prof. Dr. med. Dan Bieler, LtCol MC

---

## [Author Response · Author response to Decision Letter 0]

8 Jun 2023

GENERAL COMMENTS:

- First, although the term “trauma recidivism” is common, I would caution the authors against its use. The term “recidivism” is strongly associated with criminality and implies fault on the part of the patient. Patients injured due to violence frequently come from marginalized groups, and this language risks reinforcing racist assumptions about this patient population. Recurrent violent injury is a clear phrase that can be substituted. I recommend authors to use “recurrent traumatic injury” instead.

We thank you for this comment. Following the editor’s recommendation, the term “trauma recidivism” has been replaced with recurrent penetrating injury (RPI) throughout the manuscript.

- Please follow the STROBE guidelines style, especially for the methods section

ABSTRACT:

- Results: please provide numerical results

Thank you for this point. We have updated results section of the abstract with numerical results. 

INTRODUCTION:

- Introduction, line 1: What does “urban trauma” mean? Is this study limited to urban areas?

- I recommend that the aim of the study be explicitly stated at the end of the introduction.

We appreciate this comment. The word “urban trauma” has been replaced with “traumatic injuries” to minimize confusions, as this study was not limited to urban areas. Last two sentences of the Introduction has been revised per recommendation: “Thus, the present study examined national trends in RPI as well as factors associated with it. The aim of this study was to elucidate clinical and financial outcomes following RPI using a nationally representative cohort” (Page 4).

METHODS

- How were new injuries distinguished from patients readmitted due to sequelae or complications of their index admission?

- How was the index admission identified? Could patients have been injured prior to the first visit that is denoted here?

- The study uses ICD-9/10 codes to identify firearm- and stab-related trauma, which may not capture all cases or accurately differentiate between intentional and unintentional injuries. How did authors handle this?

- Proportion of missing data, figure 1, supplemental Table 1 are results and should not be mentioned in the methods section but in the results section instead.

We thank you for your valuable suggestions. Although ICD-10 codes differentiates between initial, subsequent and sequelae of given diagnosis, we are unable to clearly delineate whether patients were readmitted due to sequelae or complications of their index admission. We have noted the limitation as follows: “Due to our reliance on ICD-10 diagnosis and coding practices at various institutions to define penetrating injuries, patients readmitted due to sequelae or complications of their index admissions could not be clearly delineated” (Page 12). 

As noted on our Supplemental Table S1, only codes for initial encounter of injuries were used to determine index admission. 

Both the External Cause of Injury and International Classification of Diseases, Tenth Revision codes have enough granularities to show whether the injuries were intentional or non-intentional. Intentions of the penetrating injuries are noted on Table 2 and Supplemental Table S2. 

Proportion of missing data, figure 1 and supplemental Table 1 have been removed from the Method section as recommended. 

RESULTS

- Figure 1: please indicate how many patients were excluded for each exclusion criteria.

- Please provide n and % in tables not only %

- Please indicate the evolution of % GSW between 2010 and 2019

- Paragraph 3.3: Of an estimated 687,863 patients, 15,142 (2.2%) were (and not was) identified.

Thank you for this comment on the result section. We have revised the Figure 1 and Table 1 as advised. Evolution of recurrent GSW as well as percent increase in recurrent GSW hospitalizations were noted in the Result section (Page 7). Paragraph 3.3 was edited as recommended. 

DISCUSSION

- Discussion, line 1: “Penetrating trauma is the strongest predictor of future traumatic injury.” This is an often-repeated saying, but to my knowledge it has not been demonstrated. The cited reference here simply refers to Sims’ 1989 paper, which does not establish this relationship.

We appreciate your insightful point. First sentence of the discussion has been modified as the following: “Penetrating trauma has been associated with recurrence and is responsible for >$25 billion in annual costs based on medical expenses and lost productivity in the US.” (Page 9).

- By using the NRD, this analysis is limited to injuries that result in hospitalization. This is less than half of firearm injuries, with the majority being treated and released from the ED. If this data source is to be used, all of the results and conclusions must be interpreted in this context.

We thank you for this comment. We have updated the definition of our cohort as the following in the Method section: “Recurrent Penetrating Injury was defined as a subsequent penetrating injury requiring readmission within 60 days of discharge from the index hospitalization.” (Page 5). Furthermore, we have updated our result section to emphasize that our cohort definition of hospitalization due to recurrent penetrating injury with following sentences below: 

“From 2010 to 2019, hospitalizations due to gunshot wound injury increased by 138% and recurrent GSW by 196% in annual incidence (nptrend<0.001, Figure 2).” (Page 7).

“Over the study period, there was a stable trend in hospitalization due to recurrent stab injury (nptrend=0.60, Figure 2).” (Page 8).

Furthermore, in order to emphasize our limitation regarding those treated and released from the ED, we added the following sentence to the limitation section: “Despite the strength of NRD to track repeat hospitalizations at participating hospitals, information regarding the outpatient treatment of RPI as well as out-of-hospital mortality are unable to be determined.”

- I believe the NRD does not include race. Given the role of structural racism in generating risk for violent injury, this is a limitation that should be mentioned.

Thank you for this comment. The limitation section of the discussion has been edited with the following statement: “Additionally, the NRD lacks patient-identifying variables, such as race, location, and hospital data, as well as surgeon-specific variables.” (Page 12). 

- 30% of readmissions were very soon after discharge. Again, this makes me suspect that these are either transfers or complications, rather than real new injuries. Please clarify this point

Thank you for this point. Indeed, days to readmissions have been reported to be very soon after discharge since our analysis of timing was limited by the database. As noted in our Method section, the NRD tracks readmissions “within each calendar year.” Readmissions across the years were unascertainable. As we acknowledge these timing to readmissions could be misleading, we have removed them from our results section. Additionally, as we rely on ICD coding for our definition of recurrent penetrating injury cohort, our limitation section has been updated accordingly to account for any possible misattribution.

- Patients discharged to rehab, etc, had higher risk of readmission. Is it really plausible that these were new injuries? Patients with injuries that are severe enough to require rehab are less likely to be out and about. And indeed, since much of the 60 day follow up period would be spent in rehab, I suspect this may be misattribution.

We appreciate your comment. As shown in Table 3, for those discharged to skilled nursing facilities and home health, only those with GSW have been shown to have a higher risk of readmission within 60 days. This may be due to the misattribution from patients undergoing sequelae events or complications, and this has now been noted in the limitations of our manuscript. Nonetheless, our data showed that, among those hospitalized for both recurrent gunshot and stab injuries, transfer to a short-term care facility and leaving against medical advice are associated with an increased risk of readmission. Leaving against medical advice and transferring to a short-term medical facility may limit appropriate follow-up care and contribute to new injuries. 

- The conclusion of the study should be appropriately restrained and aligned with its objective

Thank you for this comment. We have revised our conclusion as the following: “The rate of RPI among patients with penetrating trauma has increased over the past decade and continues to pose a clinical and financial burden in the US. Our study underscores the contributing factors underlying repetitive incidences of gunshot and stab injuries. In light of these findings, national efforts to improve post-discharge preventive measures and bolster social support services for patients with penetrating trauma are warranted. To effectively address the mounting burden of RPI, multidisciplinary approaches involving physicians, public health leaders, and legislators should be adopted to mitigate the incidence of penetrating trauma within our communities.” (Page 12-13).

---

## [Decision Letter · Decision Letter 1]

22 Aug 2023

National Trends and Resource Utilization Associated with Recurrent Penetrating Injury

PONE-D-23-00328R1

Dear Dr. Benharash,

We’re pleased to inform you that your manuscript has been judged scientifically suitable for publication and will be formally accepted for publication once it meets all outstanding technical requirements.

Kind regards,

Francesco De Micco, M.D., Ph.D.

Academic Editor

PLOS ONE

Additional Editor Comments (optional):

Reviewers' comments:

Reviewer's Responses to Questions

**Comments to the Author**

1. If the authors have adequately addressed your comments raised in a previous round of review and you feel that this manuscript is now acceptable for publication, you may indicate that here to bypass the “Comments to the Author” section, enter your conflict of interest statement in the “Confidential to Editor” section, and submit your "Accept" recommendation.

Reviewer #1: All comments have been addressed

Reviewer #3: All comments have been addressed

2. Is the manuscript technically sound, and do the data support the conclusions?

Reviewer #1: Yes

Reviewer #3: Yes

3. Has the statistical analysis been performed appropriately and rigorously? 

Reviewer #1: Yes

Reviewer #3: Yes

4. Have the authors made all data underlying the findings in their manuscript fully available?

Reviewer #1: No

Reviewer #3: Yes

5. Is the manuscript presented in an intelligible fashion and written in standard English?

Reviewer #1: Yes

Reviewer #3: Yes

6. Review Comments to the Author

Reviewer #1: Well revised within the confines of data limitations. I don't have any additional feedback at this time.

Reviewer #3: Thank you again for the opportunity to review the manuscript.

All comments are addressed.

No further comments.

7. PLOS authors have the option to publish the peer review history of their article (what does this mean?). If published, this will include your full peer review and any attached files.

Reviewer #1: No

Reviewer #3: **Yes: **Ass. Prof. Dr. med. Dan Bieler

---

## [Editor Report · Acceptance letter]

24 Aug 2023

PONE-D-23-00328R1 

National Trends and Resource Associated with Recurrent Penetrating Injury 

Dear Dr. Benharash:

I'm pleased to inform you that your manuscript has been deemed suitable for publication in PLOS ONE. Congratulations! Your manuscript is now with our production department. 

Kind regards, 

on behalf of

Dr. Francesco De Micco 

Academic Editor

PLOS ONE